# Multi-Sensor Accurate Forklift Location and Tracking Simulation in Industrial Indoor Environments

**Valentín Barral** 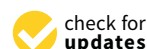**, Pedro Suárez-Casal** , **Carlos J. Escudero** and **José A. García-Naya \***

Universidade da Coruña (University of A Coruña), Centre for ICT Research (CITIC), Campus de Elviña s/n, 15017 A Coruña, Spain; valentin.barral@udc.es (V.B.); pedro.scasal@udc.es (P.S.-C.); escudero@udc.es (C.J.E.)

**\*** Correspondence: jagarcia@udc.es; Tel.: +34-881-016-086

**Abstract:** Location and tracking needs are becoming more prominent in industrial environments nowadays. Process optimization, traceability or safety are some of the topics where a positioning system can operate to improve and increase the productivity of a factory or warehouse. Among the different options, solutions based on ultra-wideband (UWB) have emerged during recent years as a good choice to obtain highly accurate estimations in indoor scenarios. However, the typical harsh wireless channel conditions found inside industrial environments, together with interferences caused by workers and machinery, constitute a challenge for this kind of system. This paper describes a real industrial problem (location and tracking of forklift trucks) that requires precise internal positioning and presents a study on the feasibility of meeting this challenge using UWB technology. To this end, a simulator of this technology was created based on UWB measurements from a set of real sensors. This simulator was used together with a location algorithm and a physical model of the forklift to obtain estimations of position in different scenarios with different obstacles. Together with the simulated UWB sensor, an additional inertial sensor and optical sensor were modeled in order to test its effect on supporting the location based on UWB. All the software created for this work is published under an open-source license and is publicly available.

**Keywords:** indoor positioning; ultra-wideband; industrial environment; industry 4.0

---

## 1. Introduction

Indoor positioning has become a hot topic in the literature because it is still quite open and subject to research and innovation, leading to many different positioning systems based on a plethora of technologies [1]. Among the considered technologies, those based on radio frequency (RF) signals have gained most of the attention because of their flexibility to be adapted to different requirements and scenarios. One of these scenarios, where indoor location systems have revealed themselves as an important source of valuable information, is the industrial environment: automation processes, optimization of operations, traceability, and many other tasks have improved because of the availability of positioning data. With the advent of the so-called industry 4.0, the need for involving positioning systems into companies' processes has become more pressing [2].

Many RF technologies are considered in the literature for indoor positioning due to a number of reasons, such as their low cost or their availability in consumer devices. However, not all of them are suitable for indoor positioning [3], mainly because of their inability to provide accurate location data in industrial environments where the presence of electromagnetic interferences is higher than in a typical domestic scenario. Examples of these technologies are Bluetooth, RFID, ZigBee, and WiFi [4,5], usually limited to secondary/low-accuracy location systems in industrial environments [6].

Ultra-wideband (UWB) excels among the very few RF technologies available when targeting the positioning of a mobile object within the three-dimensional space and with high accuracy (sub-meter

level) [7–9]. When considering UWB, several different options arise for its deployment. They are all based on the idea that the wide bandwidth of the UWB signal leads to an excellent time resolution (in the order of picoseconds), which translates to a position with centimeter-level accuracy. Among the different positioning strategies based on UWB, the so-called two-way ranging, which is based on time of arrival (TOA), does not require to synchronize the clocks of all involved nodes, leading to a deployment cost much lower than that required by other alternatives such as those based on time-difference of arrival (TDOA) [10]. Unfortunately, the performance of TOA-based UWB ranging systems is very dependent on the direct path signal impinging the receiver [10–12]. Particularly, in indoor non-line-of-sight (NLOS) conditions, the estimation of the direct signal path becomes a challenge due to the difficulties found at the time of mitigating the multipath effects [10,12,13], which are responsible for corrupting the ranging estimates with large positive biases.

In order to overcome the limitations exhibited by RF technologies when considered for positioning systems, a frequent strategy consists of incorporating additional information provided by other technologies, such as inertial (accelerometers, gyroscopes) and physical sensors (magnetometer, barometer, etc.) [8,14,15]. This so-called multi-sensor approach is introduced for the case of industrial environments, in Section 1.1.

*1.1. Positioning Systems in Industrial Environments: A Multi-Sensor Approach*

A typical workflow in a factory requires moving things around while new products are being manufactured, which are also moved to the places where they are stored before their departure. Different strategies are followed to move things depending on their size, volume, weight, etc. Among these strategies, carrying materials on forklifts is a common approach. Knowing the position of the products and materials enables for optimizing the workflow, improving the working process, reducing costs, and increasing safety. Traceability is also crucial in many cases, including the path followed by the products from the beginning to the end of their journey inside a factory. In many scenarios, locating the pallets where the products or materials are placed gives the necessary information to cover all these needs.

Nowadays, there are several commercial solutions which address the pallet tracking problem. Companies such as Vero Solutions [16] or Tech Solutions [17] rely on WiFi, RFID, and optical sensors (cameras) to track the pallets and the assets in a warehouse. Their approach involves RFID readers on the forklifts and a fixed load/unload point with more readers to track the resources once they leave the factory. Other solutions, such as the one proposed by Touchpath [18], involve also RFID readers, but in this case, they are placed in the ceiling.

All these systems, however, exhibit important limitations. Most of them need to modify the internal work-flow of the factory (pallets must be picked up or left in a specific area, readers must be placed on doors or corridors, etc). Others have a high cost if the work area is large (expensive long-distance RFID readers) or the amount of items to be tracked is also huge (one or more tags are needed on every single tracked object). Thus, the location provided by these systems is limited to some areas of interest while the rest of the facility/warehouse remains with no tracking or with a very low accurate one.

Even in small or medium-sized factories, the number of pallets to be located can be enormous, hence locating each of them with high accuracy becomes an arduous task, especially if a lower budget is present. Solutions based on including electronic tags on each pallet are unapproachable for most of the companies, and the alternatives based on the so-called computer vision typically require deploying many sensors to cover the whole factory and suffer from privacy issues.

In general, the following problems have to be addressed when deploying a positioning system in an industrial facility:

- Multipath effectscaused by reflections on metallic materials abundantly found in the infrastructure, machinery, forklifts, tools, raw material, etc. Hence, RF-based location technologies will

experience a significant performance degradation in these scenarios due to rich multipath wireless channels.

- Obstacles in an industrial environment are typically produced by the movement of people and objects, yielding a phenomena known as shadowing, which hinders the visibility of the location target.
- Operational continuity of the location process in an industrial scenario (or in a generic workplace) in the sense of producing the minimum impact on existing workflow and protocols.

Given the above-mentioned restrictions imposed by typical industrial scenarios, a single-technology approach is discarded right from the problem analysis. Instead, a combination of several sensors from different technologies is selected as the most promising solution. The simultaneous utilization of multiple sensors (chosen according to the requirements) offers several advantages compared to the single-technology solution:

- Strength: each technology has different weak points. A combination of technologies can help to overcome this problem because not all of them are going to fail simultaneously.
- Flexibility: future requirements or problems can be solved or mitigated by incorporating additional sensors.
- Granularity: Typically, different requirements are identified depending on the area or zone inside a building or warehouse. A multi-technology approach addresses better this situation by using a different set of sensors (varying its number and type) according to the needs imposed by each area.
- Cost: a higher number of technologies allows for selecting the sensors and their development tools and platforms from a larger set, thus easing the problem of finding an alternative that fits the budget.

However, considering a combination of multiple sensors to address the location problem in an industrial environment leads to a new problem: to figure out what is the combination of available sensors yielding the best performance in terms of location accuracy.

### 1.2. The Proposed Solution: Multi-Sensor Forklift Location and Tracking

Instead of including a sensor on every single product item, platform or pallet to be tracked, we propose to locate and track the forklifts. In a further step, the products or pallets can be located by identifying where they are left and picked up. The main idea behind this approach consists of locating the forklifts with the maximum accuracy (estimating not only the $x$ and $y$ coordinates, but also the forklift orientation) and, additionally, in identifying the pallets from the forklift location when they are loaded and unloaded. This approach provides significant benefits:

- The number of forklifts is orders of magnitude lower than the number of pallets, hence reducing the deployment cost.
- Maintenance costs are also lower. These costs can be crucial in the long term since the difference between maintaining only a few units in the face of hundreds or thousands is very significant.
- Different technologies can be used to perform the two above-mentioned pallet-positioning steps, and the system could be improved in the future by adding new ones.
- Forklift tracking provides value-added information and companies can exploit it profitably (e.g., route optimization and safety).

### 1.3. Main Contributions

The main contribution of this work is a realistic software simulation tool based on the Gazebo physics simulator [19] consisting of:

- A UWB simulator capable of replicating in Gazebo the behavior of this technology when used for a real-world location. Unlike other more focused approaches in replicating the radio signal at low level [20], the solution implemented in this work models the high level behavior of the range and received signal strength (RSS) values produced by this type of devices.

- A model of a vehicle (the forklift) with the motion model of this type and machines and taking into account the placement of the considered sensors on the forklift.
- A location algorithm based on Kalman filter designed to merge all the data coming from the sensors. Such a location algorithm is implemented using the robot operating system (ROS) ecosystem [21], an open source framework for writing robot software.

The proposed software environment offers the following advantages:

- The simulated vehicle (a forklift in our case) has exactly the same elements (sensors, processors, etc.) as those a real vehicle would equip.
- The location algorithm does not have to be re-implemented for the simulator. Instead, the same code that will run in the hardware is used by the simulator. In this case, the location algorithm is implemented in ROS, a well known and widely used technology.
- Multiple location algorithm implementations can be tested under the same conditions without requiring costly and time-consuming measurement campaigns.
- Additional sensors can be included in the considered sensor set to evaluate their impact on the performance of the location algorithms. Although validating the models for these new sensors might require measurements, they will not involve the whole system.
- More scenarios and vehicles can be included in the simulator to assess more complex environments.
- All the source code of the developed system [22–29] is publicly available.

The rest of the paper is structured as follows. Section 2 describes the set of low-cost off-the-shelf sensors suitable for locating forklifts in an industrial environment. Section 3 details the fundamentals of the new UWB simulator created in this work. The considered location algorithm and the corresponding motion model are detailed in Section 4. All the details of the software implementation are described in Section 5. The elements and models participating in the experiments are described in Section 6, and the simulation results are included in Section 7. Finally, Section 8 is devoted to the conclusions.

## 2. Sensor Considered for Industrial Environments

There are several different technologies in the field of real-time location system (RTLS). Among all of them, nowadays the solutions based on UWB sensors are the most valuable choices to obtain high accuracy at a moderate cost. UWB devices use TOA or TDOA as the sensing parameter, achieving a better accuracy than the technologies based on RSS (such as Bluetooth, RFID, and WiFi), which were traditionally used in indoor location systems [30]. One of the most widespread UWB integrated circuit (IC) used in industry is the Decawave DW1000 module [31], which is frequently found in many solutions such as the set of tags and anchors from Pozyx [32].

A Pozyx tag/anchor is an Arduino shield with several integrated sensors: a magnetometer, a gyroscope, an accelerometer, and the Decawave DW1000 IC. This module is an IEEE 802.15.4-2011 UWB-compliant wireless transceiver with the following features (according to the manufacturer):

- Precision up to 10 cm.
- High data rate communications, up to 6.8 byte/s.
- Highly immune to multi-path fading.
- Low power consumption.

However, some of these points were assessed by the authors in [33] and results showed slight differences with respect to the manufacturer claims. More specifically, the DW1000 IC provided highly accurate ranging measurements in scenarios where the typical RF signal propagation problems (NLOS, fading, etc.) are not very intense. However, in an industrial environment with many obstacles (workers moving around, machinery, raw material, etc.), the results provided by the DW1000 IC are not satisfactory [34,35]. NLOS propagation conditions (reflections, rebounds, etc.) between tag and anchors degrade the quality of the measurements and lead to imprecise position estimations. In this

case, the additional inertial sensors available into the Pozyx tag/anchor (magnetometer, gyroscope, and accelerometer) are very handy, becoming the base of our proposed location system. Pozyx devices can be used as anchors (a beacon or reference with a known fixed position) or tags (placed on the object to be located). For location purposes, several Pozyx anchors are needed. For 2D positioning, a minimum of three anchors is required, but a higher number improves the results and becomes a requirement to cover a big area.

Another sensor named PX4 Flow was chosen, which is an optical-flow smart camera capable of operating in indoor and outdoor scenarios [36]. The PX4 Flow sensor exploits the differences found between consecutive frames together with the data obtained from inertial sensors to estimate the velocity along both $X$ and $Y$ axes. An additional acoustic sensor provides height estimates, hence the relative displacement values can be translated into real units such as meters per second (m/s). Therefore, the PX4 Flow sensor provides linear motion estimations along both $X$ and $Y$ axes, which can be easily translated to an orientation angle with respect to the $Z$ axis. The PX4 Flow also provides a quality value for each measurement based on the quality of the processed image. The light in the environment, the ground type and texture, and the speed of the target can degrade the image quality significantly, although in a different way. Thus, poor light conditions and a uniform floor can make the sensor fail, whereas a scenario well illuminated with a noisy floor (with irregularities, stains, etc.) is the best environment to get the sensor working at its maximum potential.

## 3. UWB Simulator

We have implemented an UWB simulator based on UWB measurements and consisting of two components: a Gazebo [19] plug-in to simulate an UWB tag and a set of models representing the UWB anchors. To create the simulation scenarios, the building editor available in Gazebo [37] was used. With this editor it is possible to create walls, windows and doors, as well as to define buildings with a different number of floors. The size and position of the walls is fully editable. For this work, several scenarios were defined to test the simulator under different configurations.

The main component of the UWB simulator is the Gazebo UWB plug-in. This plug-in was created to generate ranging estimations between the tag and each anchor in the scenario. To make this estimation more realistic, a set of UWB measurements was used to model its behavior. Such measurements were obtained from the measurement bank publicly available in [38], which was created by the authors from the experiment described in [39]. To fill this UWB measurement repository, a series of strategically placed Pozyx devices were employed to record measurements at different distances in three different situations (as shown in [40–42]):

- LOS: in this situation there were no obstacles between the tag and the anchor. With this configuration, the distance estimate provided by the devices was very close to the actual distance between them.
- NLOS soft: in this situation there was an obstacle between the tag and the anchor. Despite this, the devices were able to decode the correct path, hence the distance estimate was also close to the real value. However, as a result of the attenuation caused by the obstacle, the received power level was significantly lower than that in the LOS case.
- NLOS hard: in this situation there was an obstacle that totally blocked the direct line of sight between the tag and the anchor. Unlike in the NLOS soft case, the first direct path of the signal was totally blocked, hence one of the delayed paths (corresponding to signal rebounds) was decoded instead. In this case, and because the time of flight (TOF) of the signal was considerably greater than the minimum distance, the estimate provided by the UWB devices was always greater than the actual distance. In this situation, measurements were obtained with several centimeters of error, exceeding the meter in some cases.

To develop the simulator, the measurements from these three scenarios were used to train different neural networks (NNs). The final number of layers, neurons and activation functions of each NN were automatically selected by a Bayesian optimization process [43].

The first implemented NN was to estimate the offset observed between the actual distances and the estimates collected by the Pozyx devices for the LOS and NLOS soft cases. This offset is a feature of the DW1000 chip [44] used in the Pozyx devices and depends both on the distance and the received energy [33]. In the LOS and NLOS soft situations, when the actual distance between the devices was known, it was possible to train a NN so that an estimate of the error offset could be obtained. This approach, however, could not be extended to the NLOS hard case in which a distance estimate from a signal rebound causes that the actual distance traveled by the signal be unknown. That is to say, the value estimated by the devices, in this case, corresponds to the sum of all signal rebounds plus the offset caused by the DW1000 used in Pozyx devices. However, the proportion corresponding to each of these components is unknown. For the simulator, and taking into account that the NLOS hard case is very dependent on the morphology of the scenario, we decided not to use any method to estimate such an offset.

Three other NNs were developed to estimate the received power value, its variance, and the variance of the own ranging estimation for each of the three configurations contemplated. In this way, three models capable of providing the mentioned values from the distance value existing between the tag and the anchor simulator were obtained.

In addition to these models, power limits were estimated from which the devices began to exhibit an anomalous behavior, either failing to generate the samples or generating repeated values regardless of the actual distance. These limits were later used in the simulator to avoid generating values in case the power estimates do not reach these minimum values.

All of the above models and values were used to develop the Gazebo plug-in publicly available in [27]. The mode of operation of this plug-in is as follows:

1. The tag modeled with the plug-in searches the stage for the elements marked as UWB anchors.
2. For each anchor, the plug-in uses ray tracing to check if there is a direct line of sight with the tag. Then, if the power estimate is high enough, the range is considered as belonging to the LOS scenario, hence a measurement corresponding to the distance between them is generated (after applying the offset correction factor).
3. If there is an obstacle between the tag and the anchor, the thickness of the element (i.e., obstacle attenuation) is checked. If its value is below a threshold (configurable in the plug-in), then it is considered that the first path of the signal is able to pass through it. Therefore, the NLOS soft configuration is applied and the ranging value is generated. Again, if the estimated power is below the threshold, then the ranging value would not be generated.
4. If there is an obstacle between the tag and the anchor, but its thickness exceeds the level established to be considered NLOS soft, we proceed to start a search for possible bounces. In this case, and in order to keep the computational cost reduced, ray tracing is employed to look for bounces in obstacles located at the same height as the tag (i.e., walls), but not in floors or ceilings. In addition, the search is limited to a single bounce before reaching the target. In case that after this rebound there is a path free of obstacles between the tag and the anchor, it is established that the scenario corresponds to NLOS hard and both ranging and received power estimates are generated. As in the previous cases, if the received power estimate is below the threshold, the acquired information is discarded, including the ranging estimates.

The developed plug-in also publishes the current state of the anchors, marking each of them with a different color depending on whether their way to the tag is LOS, NLOS soft or NLOS hard. Figure 1 shows, in the same scenario, how the anchors change their state depending on the position of the tag. A video of this process can be found in the Supplementary Material.

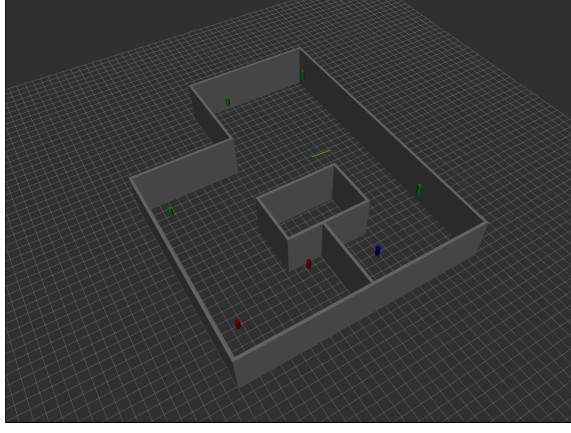 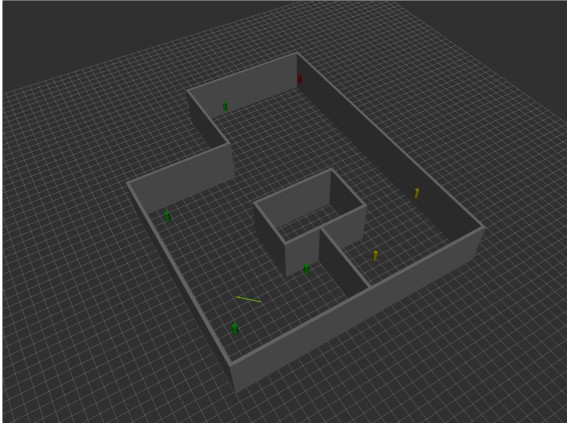

**Figure 1.** The color of the anchors indicates their state: green—line-of-sight (LOS), yellow—non-line-of-sight (NLOS) soft, blue—NLOS hard, and red—out of range.

*Simulation Accuracy*

To show the validity of the new plug-in, the original scenario was replicated in Gazebo where the UWB measurements were carried out. To do this, Gazebo's building editor was used and the walls were created according to the distances described in [39]. Figure 2 shows an image of the original scenario together with its 3D simulated equivalent. Then the three UWB anchors were placed as it was arranged during the measurement campaign, and the simulated tag (with its plug-in) was placed at the same distances as in that occasion.

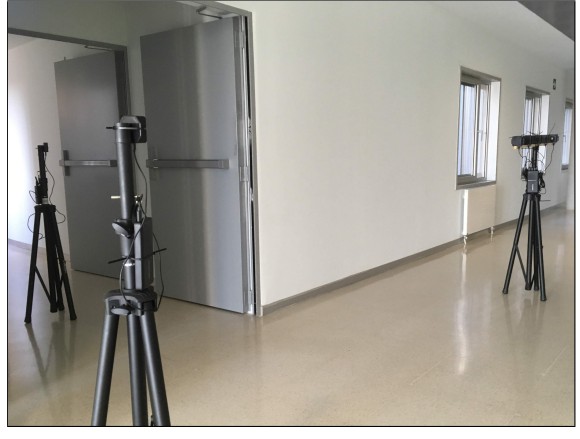 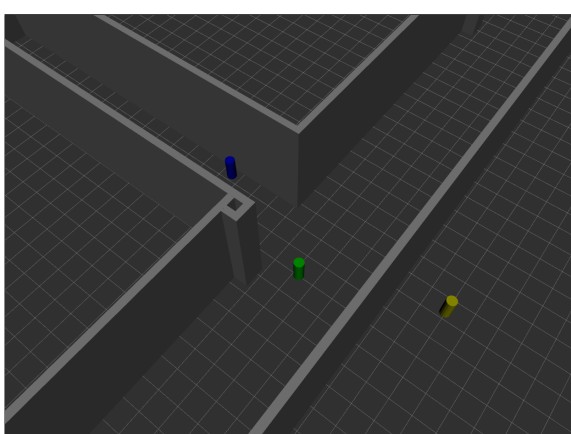

(**a**) Picture of the corridors showing two of the three anchors and the tag employed in the ultra-wideband (UWB) measurement campaign.

(**b**) Simulated scenario in Gazebo.

**Figure 2.** Measurement campaign scenario—described in [39]—and its replica in Gazebo.

Figure 3 shows the measured ranging and RSS values, whereas those provided by the UWB simulator are included in Figure 4. Figures 3 and 4 show that the simulated values approach the measured ones, especially for the LOS and NLOS soft cases, where there are no rebounds and the offset is mitigated. For the NLOS hard case the differences become larger since, in this case, the bounced signal takes different paths in both scenarios. While in the simulation is approached with a single rebound, in the actual measurement scenario the situation is much more complex and difficult to predict. However, the important key about these simulated values is that their behavior is similar—at a high level—to the measured values, since clearly the values in the NLOS hard scenario are far from the real distance values. This is what will allow us to detect later the weaknesses of a location system based only on UWB for scenarios in which LOS is not guaranteed in all points. Finally, Figure 5 plots

the mean and standard deviation of the measured and simulated ranging values, showing that the simulated LOS and NLOS soft ranging values are very close to the measured ones.

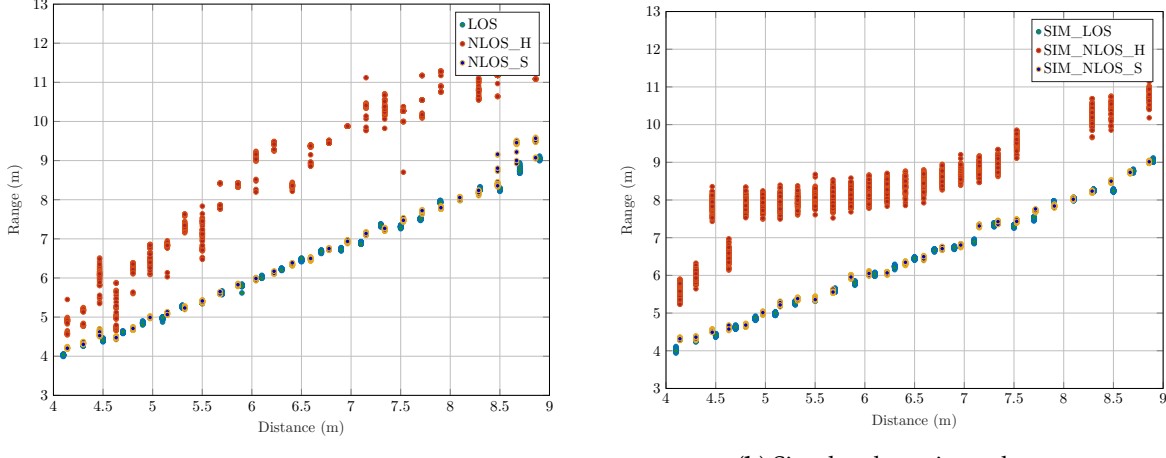

(**a**) Measured ranging values.　　　　　　　　(**b**) Simulated ranging values.

**Figure 3.** Simulated ranging values corresponding to the measurement scenario and comparison with the measured ranging values.

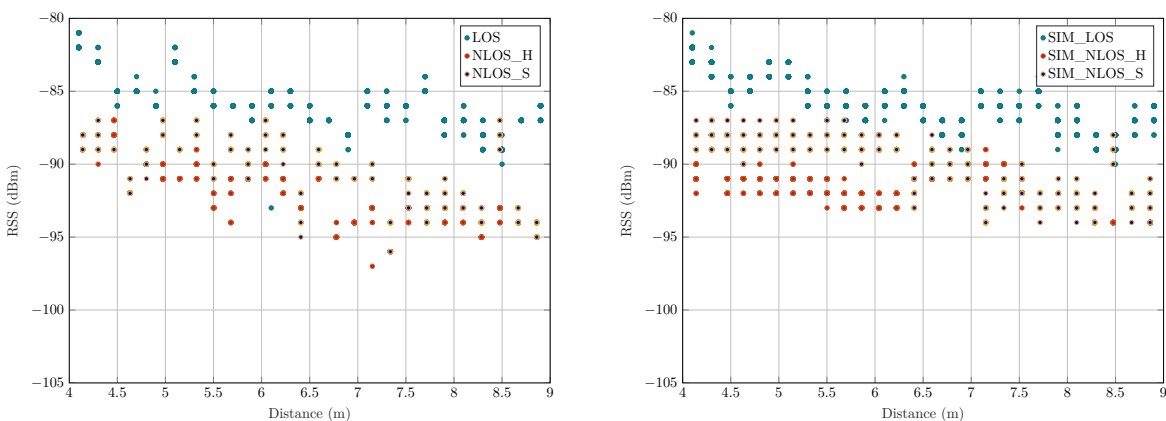

(**a**) Measured received signal strength (RSS) values.　　　(**b**) Simulated RSS values.

**Figure 4.** Simulated RSS values corresponding to the measurement scenario and comparison with the measured RSS values.

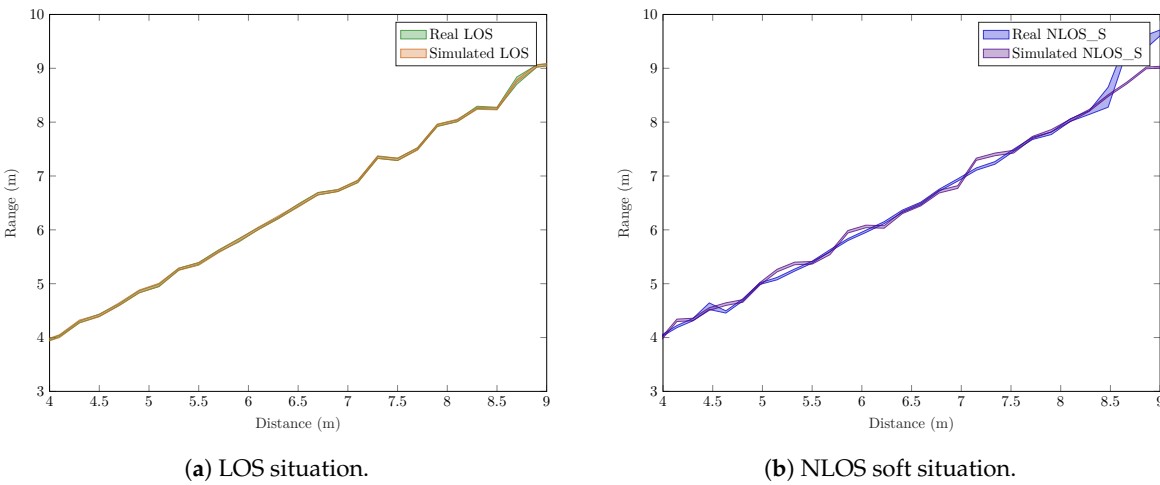

(**a**) LOS situation.　　　　　　　　(**b**) NLOS soft situation.

**Figure 5.** *Cont.*

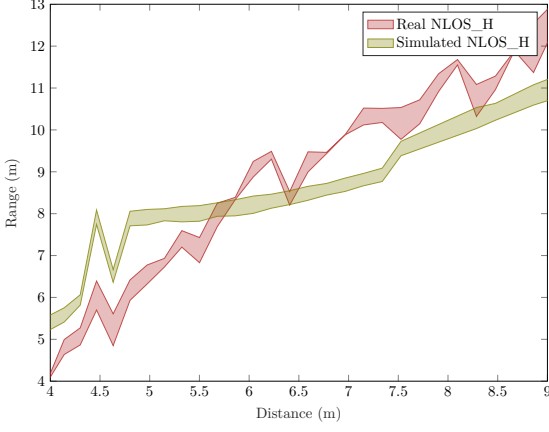

(**c**) NLOS hard situation.

**Figure 5.** Mean and standard deviation (mean $\pm\sigma$) of the measured and simulated ranging values shown in Figure 3 for the three different situations considered.

## 4. Location Algorithm

This section describes the motion model for the forklift (see Section 4.1) and the location algorithm (see Section 4.2).

### 4.1. Forklift Motion Model

The proposed forklift motion model is based on the setup in Figure 6, which shows the selected sensors placed on the roof of a forklift together with an external battery to power all the devices. The Pozyx tag is placed at some extra height over the forklift roof with a twofold objective: to avoid electromagnetic interferences from the rest of the equipment and also to maximize the LOS with the anchors. The PX4 Flow is placed on an arm on the side of the vehicle, hence the camera has full view of the ground without obstacles.

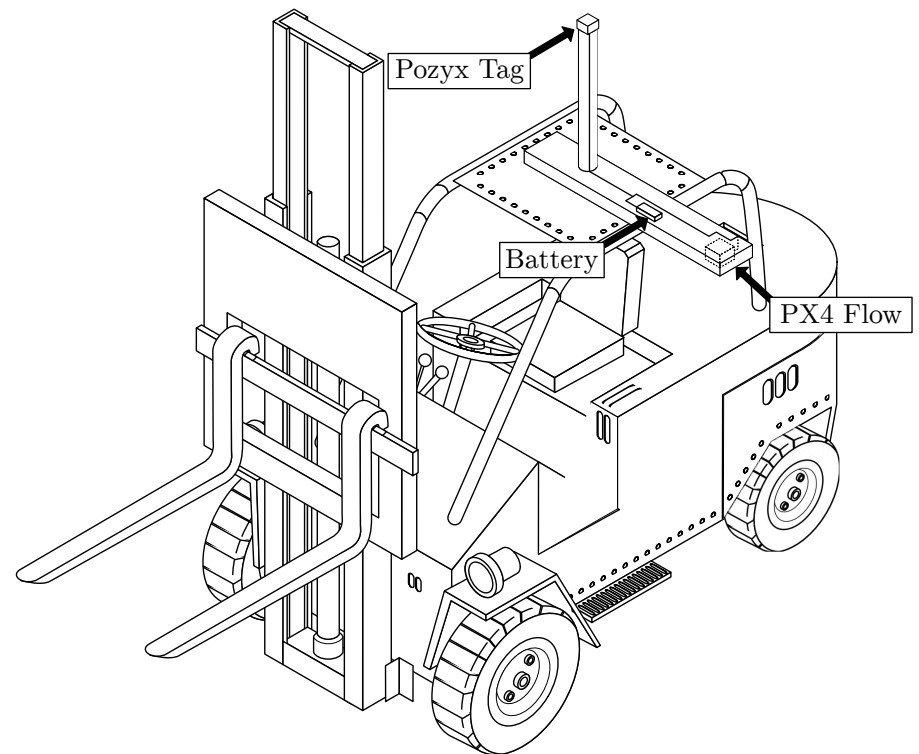

**Figure 6.** Sensor placement on the forklift.

The forklift movement model was defined taken into account that the forklift is allowed to:

- move only in two dimensions (on a plane over the ground).
- move forward and backward.
- move tracing a curve or in a straight line.
- spin around its *Z* axis without performing a displacement in *X* and *Y* axes.

*4.2. Location Algorithm Design*

The location algorithm is based on Kalman filtering, a classical approach to address a tracking problem with a recursive scheme that estimates the hidden state of a system from successive observations, and where each iteration consists of two phases. First, a prediction of the state of the system is computed from the state of the previous iteration. Next, a new estimate is obtained by combining this prediction with new information, usually coming from a set of sensors. The original Kalman filter assumes that the random variables in the system are Gaussian and that the prediction and observation functions are linear, a scenario where this filter is exact in terms of minimum mean square error (MMSE).

Many solutions have been proposed to include nonlinearities and arbitrary distributions in this scheme, such as the unscented Kalman filter (UKF), based on the unscented transform, and particle filters, which are Monte–Carlo approximations of the nonlinear system. Ranging, flow and acceleration observations are nonlinear on the pose of the tracked object, and therefore some of these solutions must be considered to obtain accurate estimations.

In this work, we have considered an iterated extended Kalman filter (IEKF) [45], which addresses this problem by linearizing these observation functions through their corresponding Jacobian matrices. In our tracking algorithm, the ranging measurements are first processed to correct the bias [33], and then these values are fed—together with the data from the other sensors—to the IEKF.

Algorithms based on Kalman filters rely on the statistical framework known as the hidden Markov model, which assumes the existence of a hidden state representing the underlying configuration of the modeled system, observed only through noisy measurements. In our case, as we are interested only in the two-dimensional location of the forklift, our state is defined as

$$\mathbf{x}_k = \left( x, y, v_x, v_y, a_x, a_y, \theta, \omega \right)^T, \tag{1}$$

where $x$ and $y$ denote the forklift position; $v_x, v_y$ its velocity; $a_x, a_y$ its acceleration; $\theta$ is its heading angle, and $\omega$ corresponds to its angular velocity.

The transition equation is another basic concept in an IEKF. This equation generates a new prediction of the state based on the current state using the rules defined in the motion model of the vehicle. In our case, the transition equation is

$$\mathbf{x}_k = \mathbf{F}\mathbf{x}_{k-1} + \mathbf{n}_k, \tag{2}$$

where $\mathbf{x}_k$ is the estimation of the forklift state, $\mathbf{n}_k \sim \mathcal{N}(0, \mathbf{C}_n)$ is a random variable modeling the prediction error, and $\mathbf{F}$ is the transition matrix defining the relationships between the state variables.

Another base concept in an IEKF is the observation equation, which relates the state of the vehicle with the data returned by the sensors. In our proposed setup, the sensors are mounted on a structure on the forklift roof as shown in Figure 7, where the Pozyx tag (A) is placed in the center of the forklift roof, whereas the PX4 Flow is located on the forklift side (B), on a sort of arm that puts the sensor outside the forklift. Taking this into account, the *observation equation* is

$$\mathbf{y}_{k,j} = H_j(\mathbf{x}_k) + \mathbf{m}_{k,j}, \tag{3}$$

where $H_j \in \{H_R, H_F, H_E\}$ represents the observation function for the ranging, flow, and accelerometers, respectively; $\mathbf{m}_{k,j} \in \{m_{k,R}, m_{k,F}, m_{k,E}\}$ corresponds to the observation noise modeled as Gaussian-distributed variables with a variance specific for each sensor type, i.e., $\mathbf{m}_{k,R} \sim \mathcal{N}(0, \mathbf{C}_{m,R})$, $\mathbf{m}_{k,F} \sim \mathcal{N}(0, \mathbf{C}_{m,F})$, and $\mathbf{m}_{k,E} \sim \mathcal{N}(0, \mathbf{C}_{m,E})$; and $\mathbf{y}_{k,j} \in \{\mathbf{y}_{k,R}, \mathbf{y}_{k,F}, \mathbf{y}_{k,E}\}$ are the corresponding observations. Therefore, we can define the corresponding observation matrices as follows:

$$H_R(\mathbf{p}) = (\|\mathbf{p} - \mathbf{b}_1\|, \dots, \|\mathbf{p} - \mathbf{b}_L\|)^T \tag{4}$$

is the observation matrix for the Pozyx sensor where $\mathbf{p} = (x, y, \bar{z})$ is the location of the vehicle, with $\bar{z}$ being the height of the tag, and $\mathbf{b}_l$ being a vector with the coordinates of the $l$-th anchor.

$$H_F(\mathbf{v}) = \begin{pmatrix} R(\theta)\mathbf{v} + \frac{1}{\Delta t}(\mathbf{I} - R(\theta))\mathbf{d} \\ \omega \end{pmatrix} \tag{5}$$

is the observation matrix for the PX4 Flow sensor where $\mathbf{v} = (v_x, v_y)$ is the velocity vector, $\mathbf{d}$ is the vector with the location of the flow sensor with respect to the center of the forklift (see Figure 7), $\omega$ is the angular velocity, and $R(\theta)$ is a standard rotation matrix defined as

$$R(\theta) = \begin{pmatrix} \cos(\theta) & \sin(\theta) \\ -\sin(\theta) & \cos(\theta) \end{pmatrix}. \tag{6}$$

Finally, the observation matrix for the accelerometer sensor is

$$H_E(\mathbf{a}) = \begin{pmatrix} R(\theta)\mathbf{a} \\ \theta \\ \omega \end{pmatrix}, \tag{7}$$

where $\mathbf{a} = (a_x, a_y)$ is the acceleration vector, $\theta$ is the heading angle, $\omega$ is the angular velocity, and $R(\theta)$ is the standard rotation matrix defined in (6).

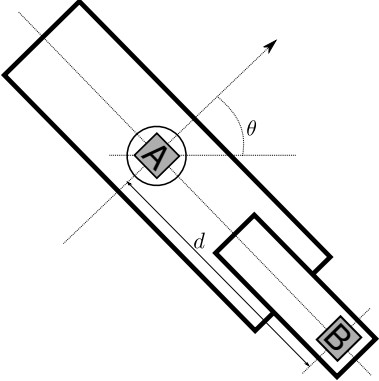

**Figure 7.** Sensor placement in detail. A: UWB tag, B: PX4 Flow.

An IEKF-based algorithm has a main loop that iterates continuously through two principal steps than run one after each other: the prediction and the update steps. The prediction step predicts the next state based on the previous one and on the motion model of the vehicle, yielding an estimation of the covariance in the prediction. In our case, it is defined as

$$\mathbf{x}_{k|k-1} = \mathbf{F}\mathbf{x}_{k-1}, \tag{8}$$
$$\mathbf{C}_{k|k-1} = \mathbf{F}\hat{\mathbf{C}}_{\mathbf{x},k-1}\mathbf{F}^T + \mathbf{C}_n, \tag{9}$$

where $\mathbf{x}_{k|k-1}$ is the predicted state and $\mathbf{C}_{k|k-1}$ is the covariance.

On the other hand, the update step compares the predicted state to the data coming from the sensors, yielding a new state estimation. In a classical Kalman filter this step is performed following the equations below:

$$\hat{\mathbf{x}}_k = \mathbf{x}_{k|k-1} + \mathbf{K}_j \left( \mathbf{y}_{k,j} - H_j \left( \mathbf{x}_{k|k-1} \right) \right), \tag{10}$$

$$\hat{\mathbf{C}}_{\mathbf{x},k} = \mathbf{C}_{k|k-1} - \mathbf{K}_j \mathbf{H}_j \mathbf{C}_{k|k-1}, \tag{11}$$

where $\mathbf{K}_j$ and $\mathbf{H}_j$ are, respectively, the linearized Kalman gain and the Jacobian matrices of Equations (4), (5) and (7) for each sensor output, i.e., ranging, flow, and accelerometers.

In an IEKF, the update step is slightly different: The update process is iterative, performing the updating task $i$ times with $i = 0, \ldots, I$, for each $k$-th time instant. Hence, the next state estimation is calculated following the equations below:

$$\mathbf{x}^0 = \mathbf{x}_{k|k-1}, \tag{12}$$

$$\mathbf{x}^{i+1} = \mathbf{x}^i + \mu \mathbf{v}^i, \quad i = 0, \ldots, I, \tag{13}$$

where

$$\begin{aligned} \mathbf{v}^i = \mathbf{x}_{k|k-1} - \mathbf{x}^i \\ + \mathbf{K}_j^i \left( \mathbf{y}_{k,j} - H_j \left( \mathbf{x}^i \right) - \mathbf{H}_j^i \left( \mathbf{x}_{k|k-1} - \mathbf{x}^i \right) \right), \end{aligned} \tag{14}$$

with $\mathbf{v}^i$ being the update direction vector in the $i$-th iteration, and $\mu \in [0, 1]$ the step size. Finally, $\mathbf{H}_j^i = \nabla H_j(\mathbf{x}^i)$ are the Jacobian matrices of Equations (4), (5) and (7) evaluated in the estimated state in each iteration, and

$$\mathbf{K}_j^i = \mathbf{C}_{k|k-1} \mathbf{H}_j^{i^T} \left( \mathbf{H}_j^i \mathbf{C}_{k|k-1} \mathbf{H}_j^{i^T} + \mathbf{C}_{m,j} \right)^{-1} \tag{15}$$

is the linearized Kalman gain with $\mathbf{K}_j \in \{\mathbf{K}_R, \mathbf{K}_F, \mathbf{K}_E\}$ the gains of the ranging, flow and accelerometer sensors, respectively. Hence, the state estimations of the forklift in the $k$-th time instant after the last iteration $I$ are

$$\hat{\mathbf{x}}_k = \mathbf{x}^I, \tag{16}$$

$$\hat{\mathbf{C}}_{\mathbf{x},k} = \left( \mathbf{I} - \mathbf{K}_j^I \mathbf{H}_j^I \right) \mathbf{C}_{k|k-1}. \tag{17}$$

The number of iterations $I$ is selected according to a cost function defined as:

$$\begin{aligned} \Gamma \left( \mathbf{x}^i \right) = \left( \mathbf{y}_{k,j} - H_j \left( \mathbf{x}^i \right) \right)^T \mathbf{C}_{m,j}^{-1} \left( \mathbf{y}_{k,j} - H_j \left( \mathbf{x}^i \right) \right) \\ + \left( \mathbf{x}_{k|k-1} - \mathbf{x}^i \right)^T \hat{\mathbf{C}}_{\mathbf{x},k}^{-1} \left( \mathbf{x}_{k|k-1} - \mathbf{x}^i \right). \end{aligned} \tag{18}$$

This $\Gamma$ function is computed after each iteration and then checked if its value falls under a threshold, in which case the loop stops. At the same time, if the $\Gamma$ function does not decrease between iterations, then the step size $\mu$ is reduced trying to ensure convergence. Regardless of the threshold mechanism, there is an additional stop condition that ends the update loop if the number of iterations overpasses a predefined limit.

## 5. Software Implementation

The location algorithm detailed in Section 4.2 was implemented in ROS, which is a widespread software environment employed in robotics-related projects due to its functionalities and helpers. It is specifically designed to interconnect different processes using a message passing mechanism,

thus each process can operate in an isolated way using publishers and subscribers to communicate with others.

Nodes and messages are the basic ROS elements. Nodes are software pieces that perform some processing and, in most cases, publish some results of interest, which are packed in the form of messages. A message is an unsorted set of fields with different identifiers and data types. A ROS node can define custom messages or use one of the generic ones offered by the ROS core. Once the result is modeled with some message type, the node publishes it under a topic identifier. Hence, several nodes can publish the same message type in different topics with distinct contents.

Besides publishing results, an ROS node can subscribe to a topic from another node and receive its data. ROS has some tools to show all the published topics in the system, their data type, and publisher settings, thus a node can find and subscribe to the desired sources. Additionally, there are several mechanisms inside ROS to record and replay the set of messages produced by different publishers, which is very useful to test and improve algorithms.

The developed ROS nodes and messages are publicly available in [22–26] and they are detailed below.

*5.1. ROS Nodes*

The following ROS nodes were developed:

- The UWB plug-in (gtec_uwb_plugin, see Table 1) publishes in the ROS ecosystem the messages corresponding to the position of the anchors in the scenario and the ranging values with respect to the tag associated to the plug-in.
- The gtec_tag_pos_publisher_plugin plug-in (see Table 2) publishes the true position of the object to which it is attached. Such position values consists of the $x$, $y$, and $z$ coordinates together with the quaternion and they are used to plot the actual trajectories followed by the object.
- The gazebo2ros node (see Table 3) acts as relay to capture the measurements of different sensors in Gazebo and publish them through ROS. More specifically, in this work it is used to publish the data from the inertial measurement unit (IMU) and the PX4 Flow sensors.
- The kfpos node (see Table 4) contains the implementation of the positioning algorithm described in Section 4, it has subscriptions to several topics (to receive data from the different sensors), and publishes the position estimation on another topic. In this specific implementation of the IEKF, the following parameters were used:

  - Maximum number of iterations: 50.
  - Minimum cost goal: $1 \times 10^{-4}$.
  - Maximum Jolt: $0.1 \, \text{m/s}^3$.

**Table 1.** Plug-in gtec_uwb_plugin . The id in the topic route corresponds to the tag identifier.

| Topic | Message Type | Topic Type |
|---|---|---|
| /gtec/gazebo/uwb/ranging/id | gtec_msgs::Ranging | Published |
| /gtec/gazebo/uwb/anchors/id | visualization_msgs::MarkerArray | Published |

**Table 2.** Plug-in gtec_tag_pos_publisher.

| Topic | Message Type | Topic Type |
|---|---|---|
| /gtec/gazebo/pos | geometry_msgs::PoseWithCovarianceStamped | Published |

**Table 3.** Node gazebo2ros.

| Topic | Message Type | Topic Type |
|---|---|---|
| /gtec/gazebo/imu | sensor_msgs::Imu | Published |
| /gtec/gazebo/PX4 Flow | mavros_msgs::OpticalFlowRad | Published |

**Table 4.** Node kfpos. The id in the ultra-wideband (UWB) range topic route corresponds to the tag identifier.

| Topic | Message Type | Topic Type |
|-------|--------------|------------|
| /gtec/gazebo/uwb/ranging/id | gtec_msgs:: Ranging | Subscription |
| /gtec/gazebo/PX4 Flow | mavros_msgs::OpticalFlowRad | Subscription |
| /gtec/gazebo/imu | sensor_msgs:: Imu | Subscription |
| /gtec/kfpos | geometry_msgs:: PoseWithCovarianceStamped | Published |

Additionally, the following external ROS nodes are also used for different purposes:

- The gazebo2rviz node [46] plots in the application *RVIZ* the position of the static elements (e.g., walls) placed in the Gazebo simulation.
- The turtlebot3_gazebo node [47] is used as a base to model the forklift vehicle.
- The teleop_twist_joy node [48] modifies the angular and linear speed of the vehicle through a USB joystick to easily move the simulated forklift across the different scenarios. To reproduce the routes, the messages of this node are recorded in a log that can be replayed later on.
- The mavros_extras node [49], which includes the definition of the message type OpticalFlowRad used to represent the output of the PX4 Flow sensor.
- The Flow_Plugin plug-in included in the PX4 Autopilot Firmware [50], which is employed to create the PX4 Flow sensor in Gazebo.

*5.2. ROS Custom Messages*

Several message types are used to model the information. Some of them are provided by the ROS core, others by the Mavros ROS package, and finally, some of them were developed ad-hoc for this project. All the custom messages were created inside a package named gtec_msgs.

The message gtec_msgs::Ranging models a generic range value between a tag and an anchor and has the following fields:

- anchorId : anchor identifier.
- tagId: tag identifier.
- range: distance value expressed in millimeters.
- seq: sequence number. Several ranging measurements can have the same sequence number if they were created at the same time.
- errorEstimation: error estimate within the provided range value.
- rss: received power estimate.

**6. Software Simulation**

Considering the Gazebo physics simulator [19], we simulate a world with a virtual forklift equipped with the selected sensors (see Section 2) placed as shown in Figures 6 and 7. Section 6.1 details some aspects about this software, whereas Section 6.2 describes the different components simulated in each of the experiments performed.

*6.1. Gazebo Physics Simulator*

Gazebo [19] is a multi-platform software consisting of several components and focused on the virtual simulation of real physical environments. Gazebo offers the following capabilities:

- A set of physics engines. Gazebo supports different physics engines such as ODE, Bullet, Simbody, or DART. They allow for simulating different physical processes such as movement, collisions, and falls.
- The OGRE 3D render engine, allowing for creating realistic 3D object models and worlds.

- Several sensor models, such as optical or motion ones, supporting noise addition to the sensor data following different models.
- A plug-in system allowing for creating new functionality and extend its base behavior, including the creation of new sensor types or new environment characteristics.

### 6.2. Simulation Elements

For the simulations we considered two different scenarios (see Figure 8 in Section 6.2.1); and a vehicle with the capabilities of a forklift equipped with UWB, IMU, and PX4 Flow sensors (see Section 6.2.2). The methodology employed to carry out the simulations and extract the corresponding results is detailed in Section 6.3.

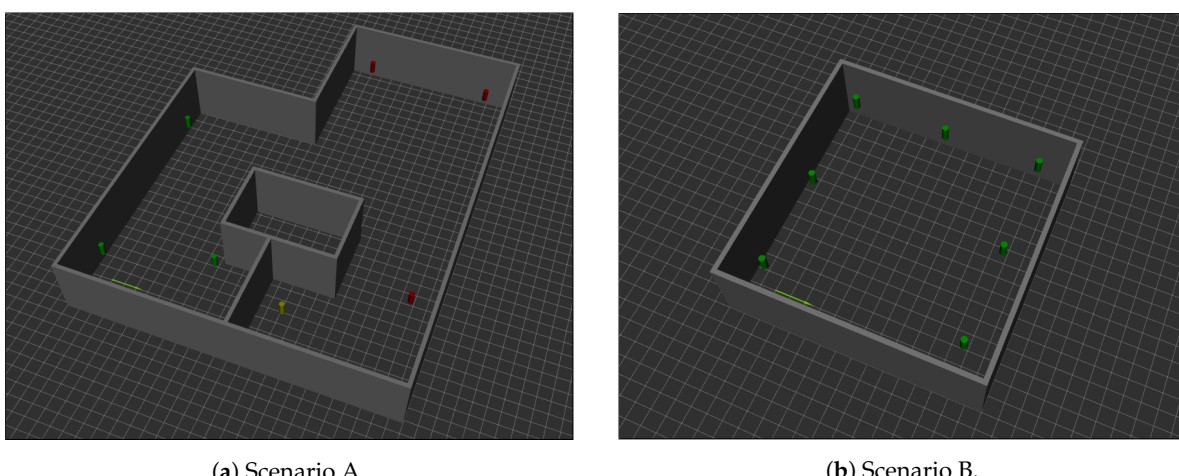

(**a**) Scenario A.　　　　　　　　　　　　　　　　　(**b**) Scenario B.

**Figure 8.** The two simulated scenarios.

### 6.2.1. Simulated Scenarios

Two different buildings were modeled for this work: scenario A, shown in Figure 8a, and scenario B, plotted in Figure 8b. In scenario A, different interior walls were placed in order to serve as obstacles between the UWB tag and the anchors in certain positions with the aim of obtaining ranging values, including values with large errors produced when some of the anchors are under NLOS hard situations. Contrarily, scenario B was designed without objects inside the movement area of the forklift with the intention of checking the performance of the positioning when a strong LOS with all UWB anchors is guaranteed.

Both scenarios were designed with a simple ground plane of 20 m × 20 m with the standard gravitational acceleration (9.8 m/s$^2$). The ground was painted with a texture to provide a visual reference for the PX4 Flow simulated sensor. This texture had to be adjusted in size and pattern draw terms to make the sensor work. Depending on the vehicle speed and the sensor height above the ground, the texture is selected in a way that the simulated PX4 Flow is able to track the movement. A texture too simple or out of focus causes the sensor to fail, as well as a texture with a pattern too regular.

### 6.2.2. Simulated Vehicle and Sensors

In order to simulate the forklift within Gazebo, the 3D mesh and sensor configuration of a conveniently scaled turtlebot3 burger robot ([47] were used. It was added an arm that protruded 1 m by one of the sides of the vehicle to place a camera along with the Flow_Plugin plug-in. A vertical pole was also placed in the center of the vehicle at a height of 1.5 m to place a cube on it simulating the UWB tag. The gtec_uwb_plugin plug-in (see Section 5) was added to this cube to provide it with the ability of producing ranging values.

The noise parameters of the IMU are specified below. These are the default parameters for the IMU sensor in Gazebo:

- Angular type: Gaussian.
- Angular $\mu = 0.0$.
- Angular $\sigma = 2 \times 10^{-4}$.
- Angular bias $\mu = 7.5 \times 10^{-6}$.
- Angular bias $\sigma = 8 \times 10^{-7}$.
- Linear type: Gaussian.
- Linear $\mu = 0.0$.
- Linear $\sigma = 1.7 \times 10^{-2}$.
- Linear bias $\mu = 0.1$.
- Linear bias $\sigma = 0.01$.

The noise parameters of the camera associated to the PX4 Flow sensor are specified below. As in the case of the IMU, these parameters are the values by default in the plugin:

- Type: Gaussian.
- $\mu = 0.0$.
- $\sigma = 1 \times 10^{-4}$.

*6.3. Simulation Methodology and Figures of Merit*

The simulation process began with the recording of two routes corresponding to both simulated scenarios. The teleop_twist_joy node [48] and a USB joystick were used, so that by moving the sticks and pressing the buttons the forklift could be guided between two points, performing different maneuvers. All the angular and linear speed change commands were recorded in a log, allowing for repeating the routes when needed.

Two additional pieces of software were developed to achieve the most accurate results: a script to re-start Gazebo simulations and record the sensor's outputs, and another different script to play a recorded log while a new instance of the location algorithm is executed. Thus, the simulation phase was split into two steps. The first step consisted of replaying the routes 100 times and record the sensor data.

After this first step, several ROS logs were recorded for each route. The second step used these logs to feed the location algorithm and to obtain the associated location estimates. The script launched each time three different configurations of the location algorithm: one using only the UWB ranging values, another using the UWB ranging values and the IMU values, and one last configuration considering all the sensors: UWB, IMU and PX4 Flow. Finally, the estimates outputted for each configuration were averaged and stored for later analysis.

All the software developed for the simulations is publicly available in [27–29].

## 7. Results

Figure 9 shows different lines highlighting the path followed by the forklift in the scenario A for each of the configurations considered for the location algorithm: using only data from the UWB sensor (dark red line), including UWB and the IMU (yellow line), and taking all the three sensors into account (purple line). The true position of the vehicle corresponds to the green line. Figure 9 shows that, when only UWB was employed, there were areas in the scenario where the error was large due to the obstacles and morphology of scenario A (see Figure 8a). Such obstacles, depending on the tag position, yield one of the following three possibilities:

1.  NLOS soft situation: the direct path can be decoded and hence the positioning algorithm provides estimates close to the true values.
2.  There was no link between the anchor and the tag, neither the direct path nor a rebound. In this situation, no ranging value wasgenerated and therefore had no influence on the algorithm.

3.   NLOS hard situation: the direct path was blocked, but a signal rebound enabled a wireless link between the anchor and the tag, although severely degrading the provided ranging estimate, hence strongly penalizing the localization algorithm.

Notice that in the experiments carried out in this work, no strategy was adopted to detect and mitigate NLOS hard situations. The authors proposed a detection and mitigation strategy in [39], which could be easily implemented in the simulator, just as it is done with the measurements. Indeed, a future line of work is to compare the performance of these NLOS classifiers and/or mitigators inside and outside the UWB simulator developed in this work.

Figure 10 shows the mean absolute error (MAE) corresponding to each of the three considered configurations of the positioning algorithm together with its 95% confidence interval. When only UWB data are considered, the average error exceeds 1 m and its variance is very large. Such high error values are due to the ranging values estimated under a NLOS *Hard* situation. Adding the inertial sensor reduces the error considerably, although its average is still above 0.8 m. Finally, adding the PX4 flow leads to the best strategy to reduce the error, which falls below 0.2 m. Figure 11 shows the empirical cumulative distribution function (ECDF) of the position error in the first scenario. Clearly, when the ranging data included values from a NLOS hard situation (and those values are not filtered out or mitigated), adding more sensors contributes to improve the behavior of the positioning algorithm. Especially notable was the improvement using the optical sensor, whose values are robust enough to keep the estimate very close to the actual position. It should be remembered that, as mentioned above, the behavior of this sensor was very dependent on the type of floor existing in the scenario, and that in the experiments carried out during this work a high quality texture was used so that its results were optimal.

Figure 12 shows the vehicle trajectories in scenario B. In this case, it can be seen how the three configurations of the positioning algorithm adopt a similar behavior. This is because in this deliberately chosen scenario, all UWB anchors were always in a LOS situation with the tag. In this way, all the ranging values generated by the simulator were very close to the true values. Consequently, the positioning algorithm worked throughout the route with information consistent with the physical reality of the vehicle and the status of its sensors.

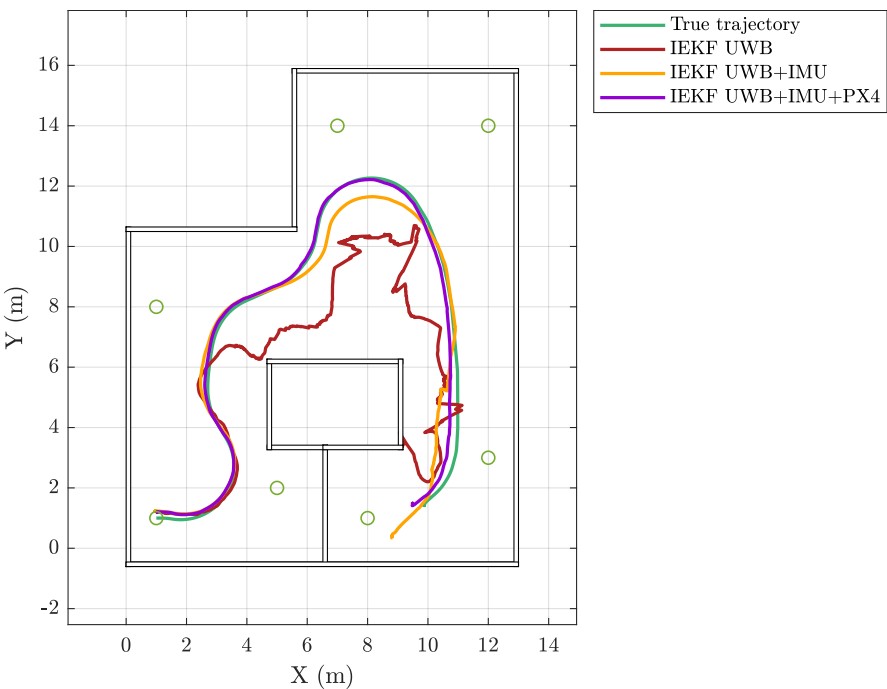

**Figure 9.** Trajectories using different sensors in scenario A.

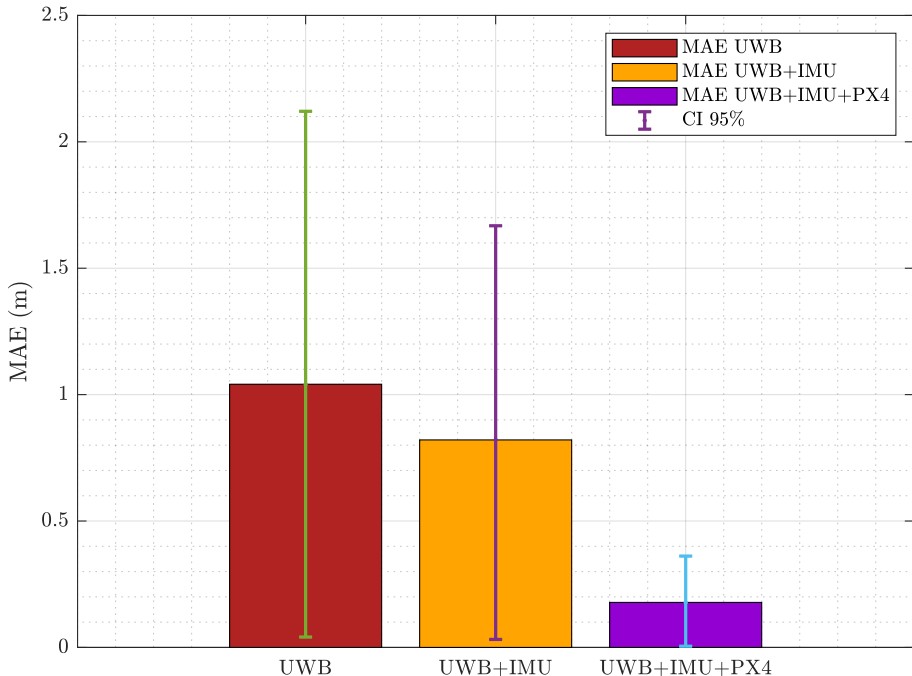

**Figure 10.** MAE of position estimation in scenario A.

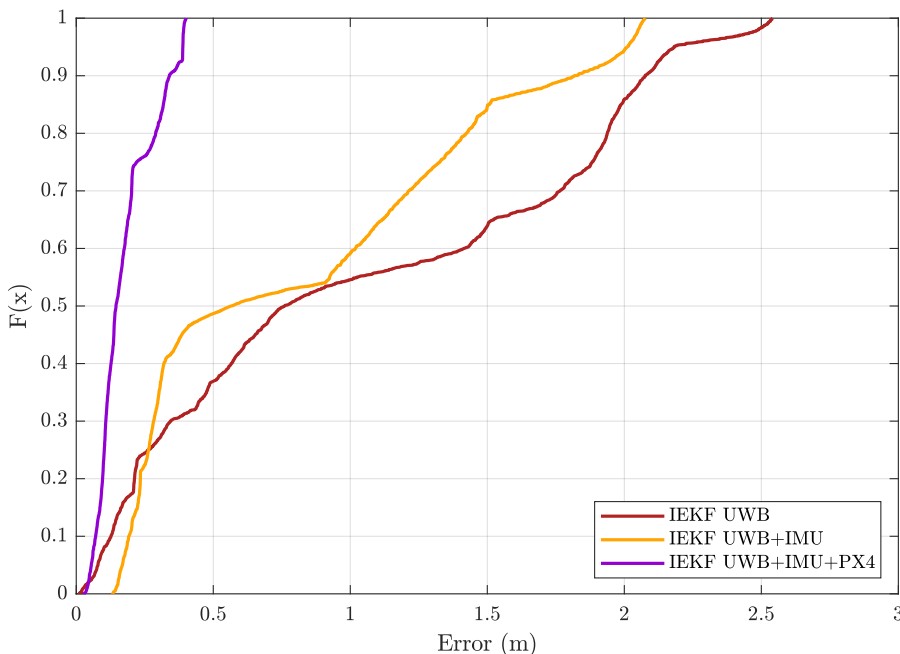

**Figure 11.** ECDF of position error in scenario A.

Finally, Figure 13 shows the MAE of the different configurations in scenario B. It can be seen how the error values are now very low, below 0.1 m even when using only the UWB sensor. Such a value corresponds exactly to the margin offered by Decawave, the manufacturer of the DW1000 base chip of the Pozyx devices, which writes in its product description *"This chip enables you to develop cost-effective RTLS solutions with precise indoor and outdoor positioning to within 10 cm."* [44]. Another important aspect of the data presented in Figure 13 is that all the configurations of the positioning algorithm perform similarly (less than 0.02 m difference between them) regardless of the number of sensors used. This indicates that probably, with the noise levels modeled on these sensors, it is not possible to significantly reduce the (already very small) error observed when using UWB measurements.

Figure 14 shows the ECDF of the position error in the same scenario. In this figure it can be seen how, although the three configurations gave a similar value of MAE, the configuration with three sensors managed to reduce the maximum error up to 13 cm, slightly below the 22 cm obtained using only UWB. It must be said, however, that with the proposed model it is not possible to infer the orientation of the vehicle using only the ranging values from a single UWB tag. For this purpose, it would be necessary to include one of the other two sensors described, either the inertial sensor or the PX4 Flow.

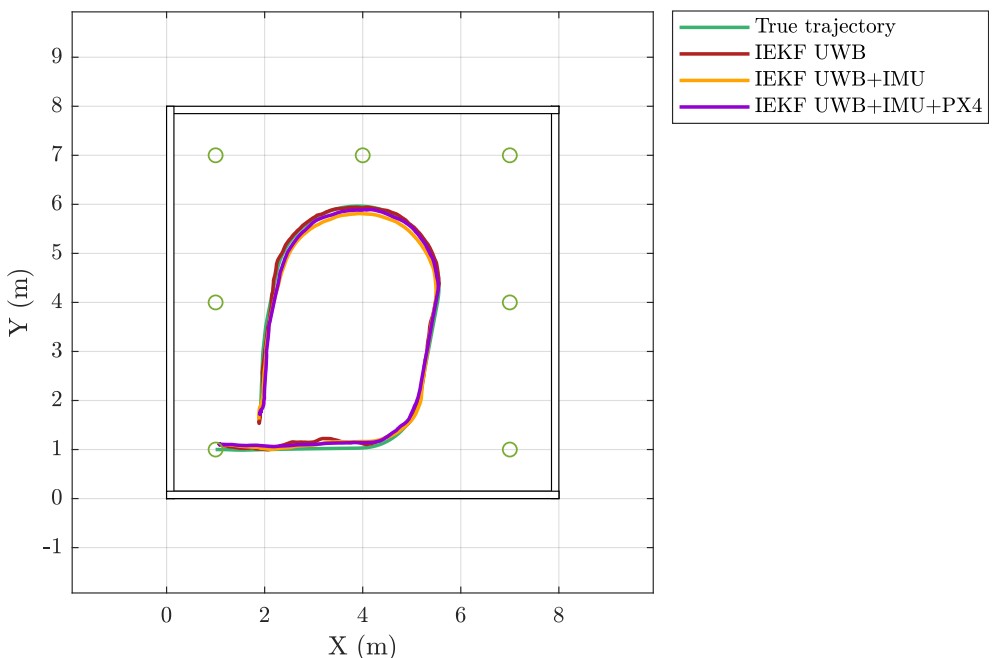

**Figure 12.** Trajectories using different sensors in scenario B.

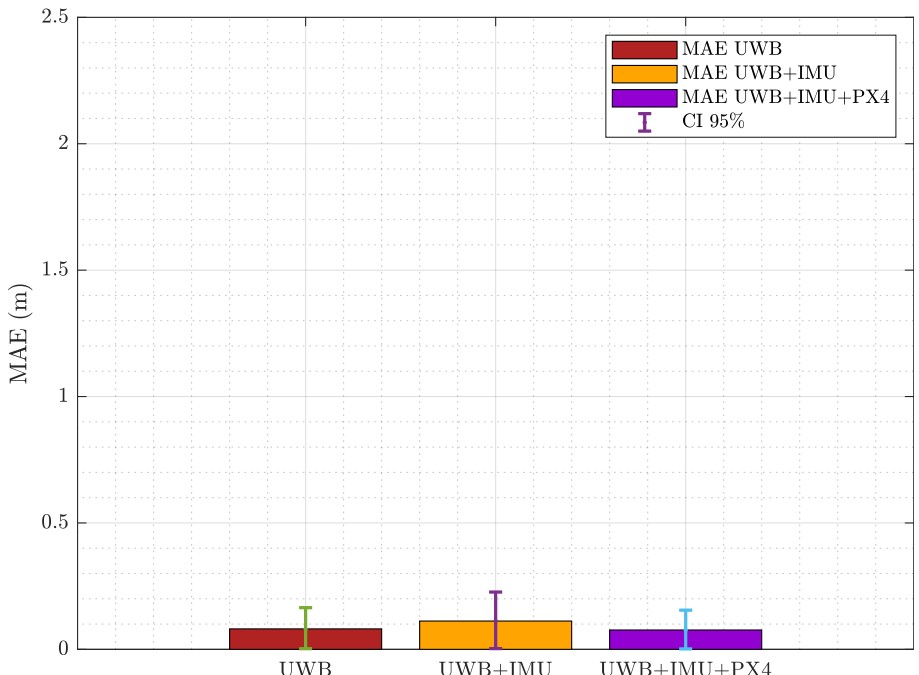

**Figure 13.** Mean absolute error (MAE) of position estimates in scenario B.

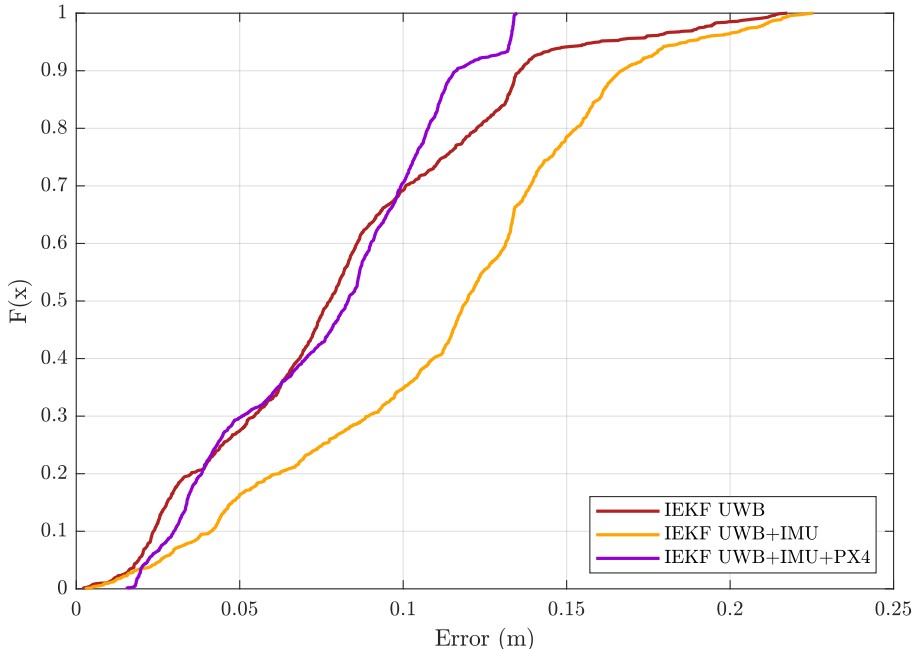

**Figure 14.** Empirical cumulative distribution function (ECDF) of position error in scenario B.

## 8. Conclusions and Future Lines

In this work, we have developed a realistic software simulation tool, based on the Gazebo physics simulator and considering multiple sensors to track forklifts in an indoor environment. We have also built a UWB simulator based on a set of measurements captured in a real scenario under three different situations: LOS, NLOS soft, and NLOS hard. To build this simulator we employed machine learning techniques to generate the different models capable of providing simulated UWB ranging values based on the distance between the tag and each anchor and the obstacles between and around them. The performance of the simulator has been tested by creating a 3D model of the place where the measurements were originally taken and comparing the simulated results with those obtained from the measurements.

We employed the considered UWB simulator together with a virtual model of the forklift and other sensors (an IMU and a PX4 Flow sensor) to provide simulated data in two different scenarios. We have used such data to feed a positioning algorithm based on the IEKF configured in three different ways: using only UWB measurements, using UWB and inertial measurements, and using data from the three sensors (UWB, IMU, and PX4 Flow).

The results of the simulations carried out revealed that the accuracy in the location using only UWB measurements is acceptable only when there is a strong LOS between the tag and each of the anchors. If this is not met and the tag receives signal bounces instead of the direct path between the two devices, then the accuracy in the location declines dramatically. The results also showed how the use of additional sensors considerably reduces the error in situations where UWB data alone are not good enough.

As future lines of work, we emphasize the capacity of the UWB simulator to support the bounces on the ground and walls and, in addition, to extend the distance limits under LOS situations to enable for simulations with larger scenarios. We will also explore the tracks opened in [39] in order to test in the UWB simulator the impact on the location when employing a classifier/mitigator capable of detecting and correcting the values produced under a NLOS hard situation.

**Supplementary Materials:** The following are available online at http://www.mdpi.com/2079-9292/8/10/1152/s1, Video S1: Comparison of real-time location accuracy using different sensor configurations: UWB only (red arrow), UWB and IMU (yellow arrow), and UWB, IMU and PX4 Flow (blue arrow). Real position corresponds to the green arrow.

**Author Contributions:** V.B. participated in the tasks: Conceptualization, formal analysis, investigation, software, visualization and writing—original draft. P.S.-C. participated in the tasks: conceptualization, formal analysis, investigation, software, visualization, and writing—original draft. C.J.E. participated in the tasks: conceptualization, funding acquisition, project administration and writing—review and editing. J.A.G.-N. participated in the tasks: conceptualization, visualization, funding acquisition, project administration and writing—review and editing.

**Funding:** This work has been funded by the Xunta de Galicia (ED431C 2016-045, ED431G/01), the Agencia Estatal de Investigación of Spain (TEC2016-75067-C4-1-R), and ERDF funds of the EU (AEI/FEDER, UE).

**Conflicts of Interest:** The authors declare no conflict of interest.

## Abbreviations

The following abbreviations are used in this manuscript:

| | |
|---|---|
| ECDF | empirical cumulative distribution function |
| IC | integrated circuit |
| IEKF | iterated extended Kalman filter |
| IMU | inertial measurement unit |
| LOS | line-of-sight |
| MAE | mean absolute error |
| MMSE | minimum mean square error |
| NN | neural network |
| NLOS | non-line-of-sight |
| RF | radio frequency |
| ROS | robot operating system |
| RSS | received signal strength |
| RTLS | real-time location system |
| TOF | time of flight |
| TOA | time of arrival |
| TDOA | time-difference of arrival |
| UKF | unscented Kalman filter |
| UWB | ultra-wideband |

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
