# Peer review of "Multi-Sensor Accurate Forklift Location and Tracking Simulation in Industrial Indoor Environments"

_electronics, doi:10.3390/electronics8101152_

Round 1

Reviewer 1 Report

The manuscript describes the results on the application of the Gazebo simular to an industrial scenario where the position of the forklifts must be estimated.

The paper is well written (even if some minor errors must be fixed) and the results are clearly presented but I have some major remarks:

1) In the introduction, the authors mention RF interference, which are also repeated in the rest of the document. On the other side, the simulation takes in consideration other aspects like fading effects and magnotometer interferences, which are not the same thing.

2) Even if the paper claims to assess the impact of distrubances in a realistic environment the simulator is used to provide the results. I do not see the effort to collect real data from a realistic scenario and give it as an input to the simulator. This is a significant limitation for a journal paper while it could be acceptable for a conference paper. I would recommend to collect realistic data from an industrial environment especially for the impact on magnetometers.

3) the IEKF has some hyperparameters to be tuned as described in the paper, but I do not see any mention of them in the results section. I imagine that they can be controlled in the Gazebo simulator.

Minor comments

page 2: Note that whay you described with Radio interferences are actually wireless propagation disturbances due to fading. Then, I propose to change the text in bold.

page 3: more than transparency is actually operational continuity or something similar.

page 4. Please insert a reference. for

UWB devices use TOA or TDOA as the sensing parameter, achieving 165 a better accuracy than the technologies based on Radio Signal Strength (RSS) (such as Bluetooth, RFID,  andWiFi), which were traditionally used in indoor location systems.

Author Response

See attached PDF containing detailed responses to Reviewers' concerns and a PDF highlighting all the changes with respect to the previous manuscript version.

Reviewer 2 Report

This paper describes a simulation study to study the positioning accuracy of a forklift using a set of different sensors Referring to positioning of anchors as shown in figure 3, the distance between the forklift and the anchors are very close, less than 4m.  It would be more realistic if the distance can be made larger The amount of noise and bias added to all sensors should be justified and tabulated for the results to be read more meaningfully (e.g. from measurements) The required accuracy for tracking the forklift should be clearly justified and stated, so that the proposed positioning solution can be assessed objectively From the results shown in figure 5 and figure 7, it seems that adding additional sensors to the UWB solution improves the positioning accuracy only marginally, in terms of centimeters (up to 10cm) RMSE, considering movement of such large vehicle as a forklift If the purpose is to track the position of the forklift, orientation tracking may not be important - unless we need to know where the forklift is facing It is mentioned that the pattern on the floor has to be carefully drawn for the optical flow sensor to work.  However in real lift the floor may be dirty and the paint work may be worn off.  How do we address this problem?

Author Response

(The authors gave the same response as above.)

Reviewer 3 Report

The authors presented a simulation of multi-sensor based forklift positioning algorithm in indoor manufacturing environments. I cannot recommend its publication based on the following issues.

1) The implemented multiple sensor fusion algorithm is based on iterated extended Kalman filter, which has been known for more than forty years. There is no major scientific/technical contribution to the field.

2) The noise models applied to all the sensors are Gaussian distributions, which is not justified.

3) The paper claims to solve positioning and tracking issues in industrial indoor environments. However, the two simulated scenarios are extremely simplified. The simulated environment is a completely vacant rectangle shaped room with no obstacles, no interferences, no multipaths, no building structures. Such kind of simulation provides very little useful information to real applications.

4) The results are poor and the explanation is lousy. The simulated performance of Scenario 8 is worse than Scenario 4, although Scenario 8 includes even one more type of sensor than Scenario 4. The authors explained that the magnetometer's covariance error is underestimated, which causes a mismatch between the expected and the actual values. I don't accept this explanation. If the measurements are taken from real sensors, a mismatch is possible. But the measurements in this paper are actually simulated, which means they are synthesized according to the model authors enforced to the simulation. There shouldn't be a mismatch. If there is one, it should be correctable.

5) If UWB sensor alone cannot provide orientation information, it's not meaningful to include it in Figure 6 and Figure 8 as a single modality. Removing it will allow better comparison of the orientation errors in other types of sensor combinations.

Author Response

(The authors gave the same response as above.)
